# Effects of Carbonated Beverage Consumption on Oral pH and Bacterial Proliferation in Adolescents: A Randomized Crossover Clinical Trial

**DOI:** 10.3390/life12111776

**Published:** 2022-11-03

**Authors:** Guadalupe Carolina Barajas-Torres, Miguel Klünder-Klünder, Juan Garduño-Espinosa, Israel Parra-Ortega, María Isabel Franco-Hernández, América Liliana Miranda-Lora

**Affiliations:** 1Departamento de Investigación Clínica, Hospital Infantil de México Federico Gómez, Secretaría de Salud, Dr. Márquez No. 162, Col Doctores, Delegación Cuauhtémoc, Mexico City 06720, Mexico; 2Departamento de Gestión de la Investigación, Hospital Infantil de México Federico Gómez, Secretaría de Salud, Dr. Márquez No. 162, Col Doctores, Delegación Cuauhtémoc, Mexico City 06720, Mexico; 3Dirección de Investigación, Hospital Infantil de México Federico Gómez, Secretaría de Salud, Dr. Márquez No. 162, Col Doctores, Delegación Cuauhtémoc, Mexico City 06720, Mexico; 4Laboratorio Clínico, Hospital Infantil de México Federico Gómez, Secretaría de Salud, Dr. Márquez No. 162, Col Doctores, Delegación Cuauhtémoc, Mexico City 06720, Mexico; 5Unidad de Investigación Epidemiológica en Endocrinología y Nutrición, Hospital Infantil de México Federico Gómez, Secretaría de Salud, Dr. Márquez No. 162, Col Doctores, Delegación Cuauhtémoc, Mexico City 06720, Mexico

**Keywords:** pH, caries, dental biofilm, carbonated beverages, saliva, artificial sweeteners

## Abstract

Sugary soft drinks modify salivary pH and favor bacterial proliferation and are associated with the development of caries. Information on the effects of consuming carbonated drinks without sucrose is limited. **Methods**: In this crossover clinical trial, salivary and dental biofilm pH were determined at 0, 5, 10, 15, 30, 45, and 60 min after the participants (n = 18) ingested a soft drink with sucrose, a soft drink with aspartame/acesulfame K, carbonated water, and plain water on different days. Dental biofilm cultures were conducted at 0- and 120-min. **Results**: Salivary pH decreased significantly after ingestion of the sucrose-containing soft drink when compared with the other types of beverages (median difference, −0.3–−0.4, *p* ≤ 0.05), and the greatest difference was found with mineral water. A greater bacterial proliferation (Colony Forming Units [CFU]) was observed after ingestion of the drink with sucrose (↑310 × 10^3^ CFU, *p* ≤ 0.01), followed by the drink with aspartame/acesulfame K (↑160 × 10^3^ CFU, *p* ≤ 0.01) and carbonated water (↑60 × 10^3^ CFU, *p* ≤ 0.05). No significant changes in bacterial proliferation were observed after the consumption of natural water. **Conclusions**: Ingestion of sucrose-containing soft drinks favors the acidification of salivary pH and the bacterial proliferation of dental biofilm. Although to a lesser extent, soft drinks containing aspartame/acesulfame K also favor bacterial proliferation.

## 1. Introduction

The World Health Organization defines dental caries as “a localized, post-eruptive, pathological process of external origin involving softening of the hard tooth structure and proceeding to the formation of a cavity” [1]. Caries is a dynamic, non-communicable disease, modulated by diet and mediated by dental biofilm [2]. It is the main dental problem worldwide and affects individuals starting in their first stages of life. Levels of prevalence have been described for the schoolchildren population around the world between 60% and 90% [3]. The prevalence among Mexican adolescents has been estimated at 72.7% [4].

Dental caries is considered a multifactorial disease that involves the following elements in its development: 1. risk factors (low income level, lower educational level, poor oral health behaviors, genetic factors, etc.); 2. frequent exposure to dietary sugars that increase the proliferation of specific dental biofilm (*Streptococcus mutans*); 3. a decrease in pH brought about by the generation of acids (lactic, propionic, and butyric acid) from bacterial metabolism (unlike dental erosion); and 4. Permanent tooth demineralization [5,6,7,8].

Tooth enamel is an acellular tissue composed of minerals (85% in volume) described as substituted calcium hydroxyapatite, water (12% by volume), and organic, usually proteins and lipids (3% by volume). The mineral component of enamel is basically calcium hydroxyapatite and the stoichiometric formula for hydroxyapatite is Ca_10_ (PO_4_)_6_ (OH)_2_ [9]. Hydroxyapatite dissolution is the chemical loss of the minerals in the dental hard tissues through the continuous decrease in pH in the oral cavity [3].

Dental biofilm is made up of billions of bacteria (the majority of which are only found in the oral cavity), growing in a mass of soluble and insoluble carbohydrates [9]. The oral bacteria *Streptococcus mutans* is generally recognized as a major etiological factor in dental caries, it resides primarily in dental biofilm on the tooth surfaces [10] and has different capabilities: 1. Acidogenicity: the ability to transport and metabolize carbohydrates into organic acids, 2. Aciduricity: the ability to thrive under low pH and 3. Polymers synthesis: the ability to synthesize extracellular polymers of glucan from sucrose which promotes adhesion to the teeth and assists colonization [9].

Therefore, the frequent intake of sucrose results in an optimal substrate for its metabolism and multiplication and creates favorable niches for the proliferation of other bacterial species [10].

Salivary pH at rest ranges from 6.2 to 7.6 [11]. Currently, the hydroxyapatite crystals dissolve minimally in saliva and have the capacity to reintegrate into tooth enamel. Pathogenic bacteria can produce acids and reduce the pH to ≤5.5 (the critical pH for dental health) after the ingestion of fermentable carbohydrates. This forms acid phosphate complexes that do not have the ability to re-incorporate into the tooth, giving rise to dental demineralization [9,12,13,14]. Finally, components are secreted through the saliva that favor the gradual return of the pH to the baseline level through inorganic elements such as bicarbonate, calcium, and phosphate ions that make up the buffer system, which limits the loss of minerals in the dental structure [15,16,17].

Saliva contributes to maintaining the oral microenvironment through the presence of numerous cationic peptides, immunoglobulin A, and salivary proteins in addition to regulating the pH [18,19,20,21]. However, these defense mechanisms may not be efficient in cases when there is frequent exposure to exogenous aggressors, such as the consumption of food o beverages high in sugar [9,22].

Non-caloric sweeteners (aspartame/acesulfame K, sucralose, Stevia©) have been incorporated into these products as an alternative to the use of sugar to sweeten soft drinks, and they produce a high sensation of the sweet taste with very low concentrations. Their consumption has increased in recent years [23]. Their incorporation into these beverages has been approved and they are considered to be safe substances for consumption [24]. However, there is limited information on their effects on oral health.

Drinks with sweeteners are known to produce a smaller drop in salivary pH when compared with sucrose-containing drinks [25,26]. However, the effects of other carbonated beverages with and without noncaloric sweeteners on both pH and bacterial growth have not been studied in depth. For this reason, our objective was to compare the effect of consuming soft drinks with sucrose, soft drinks with aspartame/acesulfame K, carbonated water, and natural water on oral pH and bacterial proliferation.

## 2. Materials and Methods

### 2.1. Methodology

This was a crossover randomized clinical trial (with the objective that each individual would be their own comparator), performed between January 2018 and February 2019 at Hospital Infantil de México Federico Gómez. Adolescents aged 12–18 years, of both sexes, who reported habitual consumption of soft drinks, and when evaluated by a pediatric dentist had a DMFT (Decayed, Missing, and Filled Teeth) index [27] of at least 3, were included. Sociodemographic data were collected, as well as information about the frequency of consumption of cola drinks and tooth brushing habits. The exclusion criteria were as follows: not undergoing orthodontic treatment, not having received a topical application of fluoride during the last 3 months, not having a motor disability that interfered with tooth brushing, not consuming drugs or being carriers of diseases that cause xerostomia, not being under antibiotic therapy during the study period, and not having active periodontal infections. All participants and tutors signed letters of assent and informed consent forms, respectively.

### 2.2. Intervention

After being selected by an investigator, participants attended four sessions with a 1-week washout interval. A standardized diet was indicated for dinner the day before each of the visits (sandwich with turkey ham and semi-skimmed milk). Subsequently, participants could ingest only plain water for up to 2 h prior to the session. They were also asked not to brush their teeth during the 48 h before the study.

All participants received 355 mL of plain water in the first session. The sequence of soft drinks for the three following appointments was randomly determined by an investigator using a computer algorithm. Then, an investigator blinded to allocation gave each participant 355 mL of one of the following types of soft drink: (1) soft drink with sucrose, (2) soft drink with aspartame/acesulfame K, and (3) carbonated water. The drinks were consumed at a temperature close to 4 °C in a maximum time of 10 min.

The decision to collect pH data at different times was based on earlier studies that reported differences caused by beverages containing caloric and non-caloric sweeteners, as well as differences in bacterial proliferation [25].

### 2.3. Outcomes

#### 2.3.1. Salivary pH

Participants were asked to spit 2 mL of saliva into a wide-mouth sterile vial at baseline and at 5, 10, 15, 30, 45, and 60 min after consuming each of the beverages. pH was measured with a HANNA HI 221 potentiometer (HANNA Instruments Inc. Woonsocket-RI-USA, Romania), with previous calibration of the electrode using buffer solutions of pH 4.0 and 7.0. The electrode was washed with distilled water between each reading.

#### 2.3.2. Dental Biofilm pH

Samples of dental biofilm were obtained using a sterile dental explorer in dental sites that represented all buccal quadrants at baseline and at 5, 10, 15, 30, 45, and 60 min after consuming each of the beverages, with a 30-s collection time. The sample was placed in 2 mL of sterile bi-distilled water and pH was determined using a potentiometer (HANNA Instruments Inc. Woonsocket, RI, USA).

#### 2.3.3. Bacterial Growth of Dental Biofilm

Dental biofilm samples were obtained using a sterile explorer from dental sites representing all buccal quadrants at baseline and 120 min after ingestion of each of the beverages. The samples were placed in BHI (Brain Heart Infusion) culture medium. A 50-microliter aliquot was deposited in 4950 microliters of physiological saline solution and placed in a vortex until the sample was homogenized. Subsequently, 100 mL of the solution was placed in plates with Todd-Hewitt culture medium and spread evenly with a sterile glass angle. The plates were incubated at 37 °C for 24–48 h in an environment supplemented with 10% CO_2_. The lyophilized strain of *Streptococcus mutans* American Type Culture Collection (ATCC) MicroBioLogics (35668) was used as a reference for comparison.

## 3. Statistical Analysis

Variables were described with frequencies and percentages or medians and interquartile range (IQR) according to the type of variable. Salivary pH and dental biofilm at different times were compared using Friedman’s analysis with adjustment for multiple comparisons using Dunn’s correction. Changes in the bacterial proliferation of the dental biofilm at baseline and at 120 min were compared using the Wilcoxon test and the drinks were compared using the Kruskal–Wallis test. The statistical program SPSS v. 22 was used and statistical significance was considered with a *p* ≤ 0.05.

## 4. Results

### 4.1. Flow Diagram

The number of participants chosen and who continued after the exclusion criteria were applied and the loss of follow-up in the study are shown in Figure 1.

### 4.2. Participants’ General Characteristics

All were consumers of soft drinks, most performed oral hygiene at least twice a day, and it was determined that all had caries lesions during the oral examination (Table 1).

### 4.3. Kinetics of Salivary pH

A significant reduction in pH can be seen during the first 5 min with a median pH of 6.80 (IQR, 6.35–7.10) after ingesting the soft drink with sucrose. Subsequently, there is a gradual recovery that reaches baseline pH levels up to 45 min.

The pH obtained after the consumption of the drink with sucrose at 5 and 10 min was significantly lower than that seen with the other drinks (*p* < 0.05). There were no significant differences in salivary pH after ingestion of soft drinks with aspartame/acesulfame K, carbonated water, and plain water, and their values remained >7.0 (Figure 2).

### 4.4. Kinetics of Dental Biofilm pH

Variability was observed in the measurements of biofilm pH kinetics without finding significant differences between the effects of the different beverages (Figure 3).

### 4.5. Bacterial Proliferation

Changes in bacterial proliferation at baseline and 120 min after the ingestion of each of the drinks (Figure 4A) and the differences between them (Figure 4B) (two extreme values were removed).

Greater bacterial proliferation was observed after consumption of the soft drink with sucrose (↑310 × 10^3^ CFU, *p* ≤ 0.001), followed by the soft drink with aspartame/acesulfame K (↑160 × 10^3^ CFU, *p* ≤ 0.001), and carbonated water (↑60 × 10^3^ CFU, *p* ≤ 0.001). There were no significant changes in bacterial proliferation after consuming natural water.

## 5. Discussion

As far as we know, this is the first clinical trial to evaluate the effect of carbonated beverages with caloric and non-caloric sweeteners in vivo on three cariogenic mechanisms (salivary pH, dental biofilm pH, and changes in bacterial proliferation). It also allows these factors to be compared with controls such as mineral water and natural water. Furthermore, this study considers the intake of usual doses of these beverages (355 mL can), which is closer to actual consumption, unlike other authors who assess exposure through rinsing with 10–15 mL [25,26] or ingesting amounts less than 100 mL [28].

We observed acidification of salivary pH within the first few minutes after exposure to the sucrose drink, as expected. This effect is similar to that reported by Sánchez et al. [28]; however, we identified lower acidification (5.8 vs. 6.8) in our study. This may be because Sánchez et al. evaluated pH levels from the first minute after ingestion, while in this study the measurements began at the 5 min mark. The combined consumption of other acidifiers in the diet can have a synergistic effect in lowering the pH even though the critical demineralization point of pH 5.5 was not reached in either study. Moreover, we observed a prolonged return of the pH to basal levels after ingesting the drink with sucrose (45 min) in our study, which allows us to estimate the time of the cariogenic potential of these drinks.

Despite the similarity between the components of the soft drink containing sucrose and that added with aspartame/acesulfame K, their effects on salivary pH were different. Therefore, the greater acidifying capacity of a regular soft drink can be attributed more to its sucrose content and not to other compounds such as phosphoric and citric acid.

We obtained results similar to those reported by Uma for mineral water, identifying a slight alkalinization [29]. However, no statistically significant differences were observed with changes in pH after consuming plain water.

The acidification time of the dental biofilm after ingesting the soft drink with sucrose was similar to that found by Brambilla, Jawale, and Ross [25,26,30], with maximum values of acidification between 5 and 15 min. However, these authors report higher levels of acidification (pH 5.14 ± 0.5, 5.72 ± 0.2, and 5.51 ± 0.5, respectively) than those determined in our study (pH, 6.80; RIC, 6.35–7.10). This may be because these authors collected samples from interdental areas, where its buffer effect is reduced by the difficult accessibility of saliva [31]. However, Saeed who also evaluated the pH of dental biofilm taken from vestibular surfaces, as in our study, found greater acidity after ingesting the drink containing sucrose (5.86 vs. 6.55 at 5 min) [32]. This is possibly because the exposure was conducted by rinsing and not after drinking the beverages, as in our study, which could have facilitated a greater impregnation of the sugars in the biofilm.

We did not find statistically significant differences in changes in the pH of dental biofilm between the different types of beverages in this study. The above differs from that reported by Brambilla, who identified greater acidification after exposure to the drink with sucrose (pH, 5.14 ± 0.05) compared with the solution with non-caloric sweeteners, such as rebaudioside A (pH, 7.11 ± 0.03) and stevioside (pH 7.06 ± 0.03). There were also differences in the type of exposure (rinses of 1 min vs. usual intake) and the concentration of the exposure substances (10% solution vs. commercial formulas) [25], even though the differences can be attributed to the type of sweeteners assessed.

In our investigation, the greatest increase in bacterial proliferation was identified after consuming a drink containing sucrose compared with non-caloric drinks. Other authors have reported that most commercial sweeteners appear to be less cariogenic than sucrose but still retain some enamel demineralization potential [33,34]. This is also consistent with what was previously reported by Brambilla [25], who identified twice the proliferation after ingesting a drink with sucrose compared with those drinks containing rebaudioside. Bacterial proliferation was also identified in beverages containing aspartame/acesulfame K, although to a lesser degree. This suggests that other compounds apart from the sweetener, such as citric and phosphoric acid, can promote the development of acidophilic microorganisms.

The carbonated beverage with which the least bacterial proliferation occurred was mineral water, and even though an increase of 60 × 10^3^ CFU was reported after consumption, this difference was not statistically different from that of plain water. Plain water was the only beverage that did not show an increase in bacterial proliferation after ingestion.

Within the limitations of this study, we recognize the small sample size as a limitation. This did not allow us to observe significant differences in changes in the pH of dental biofilm. Although each patient was his own control, it is necessary to include a larger sample size for further studies.

Furthermore, the first 5 min after ingestion could be of great relevance in the identification of changes in pH that was not analyzed in this investigation. However, we consider the isolated intake of each type of beverage in the present assay. However, we know that in daily life consumption may not be limited to 355 mL or may be accompanied by other foods and beverages that could enhance or neutralize the acidifying effects.

This study shows changes in a small group of patients with an evaluation in a single exposure time. However, these effects can have a great impact at the community level as observed in the study by Hasheminejad et al. They reported that for patients who have never consumed soft beverages it is possible to observe a DMFT index of up to 39% less than those who had had a daily intake of such beverages [35].

Therefore, questions arise for future studies to evaluate the kinetics of pH and bacterial proliferation during the first minutes of consumption, the intake of different volumes, and/or the combination with other foods and beverages.

We observed that the oral pH recovery time is prolonged (up to 45 min to return to the basal state), so recurrent consumption can cause a persistent state of acidification, predisposing to cariogenic mechanisms. This can be limited by recommending a lower consumption of these drinks and tooth brushing. It has been reported that the intake of non-sweetened beverages such as water, and more tooth brushing, are associated with fewer caries in children [36]. Oral hygiene is of such relevance that it has even been related to a greater impact on oral health than a correct diet [37].

## 6. Conclusions

The greatest salivary acidification was observed after ingesting soft drinks containing sucrose, and a prolonged recovery time to the basal state was observed although it was not documented that the decrease in pH reached the critical demineralizing point. Moreover, the ingestion of these beverages favored bacterial proliferation, which increases their cariogenic potential. It should be noted that carbonated drinks without sucrose were not inert since they also favored bacterial proliferation, although to a lesser extent. We only evaluated a single exposure time, but we would expect that repeated exposure to these beverages could have a major deleterious impact on oral health. These findings support the recommendation to discourage the consumption of carbonated beverages, mainly those with sucrose, and to encourage the consumption of plain water in the population.

## Figures and Tables

**Figure 1 life-12-01776-f001:**
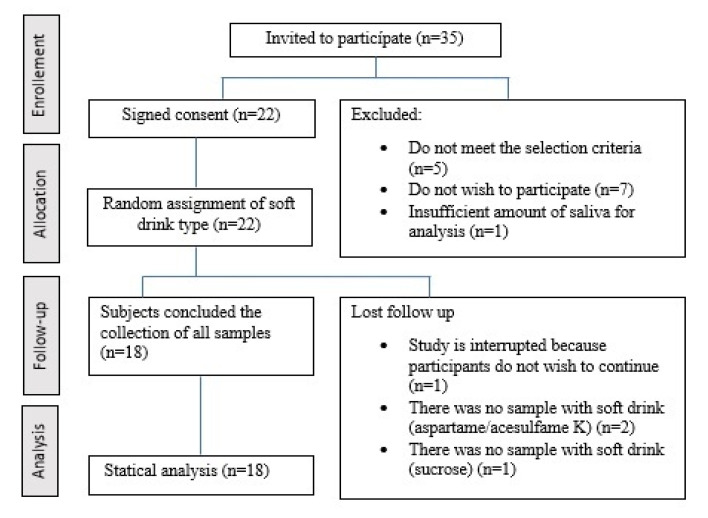
Diagram of subjects included in the study.

**Figure 2 life-12-01776-f002:**
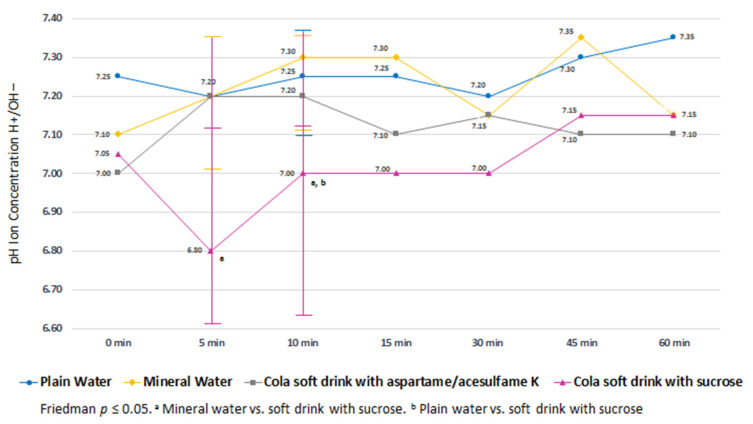
Kinetics of salivary pH after ingesting different types of beverages.

**Figure 3 life-12-01776-f003:**
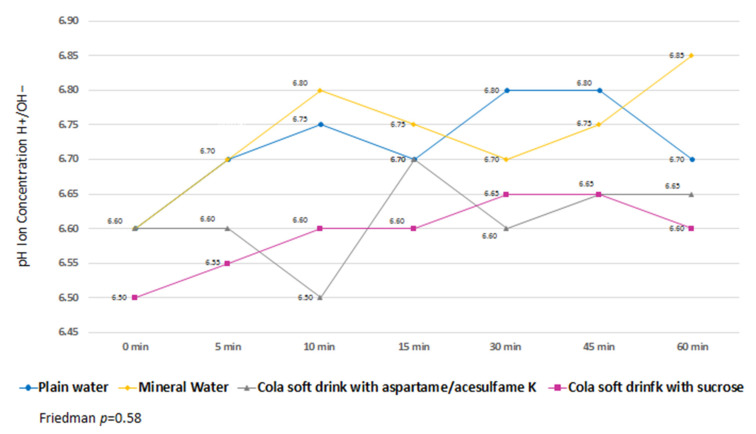
Kinetics of dental biofilm pH.

**Figure 4 life-12-01776-f004:**
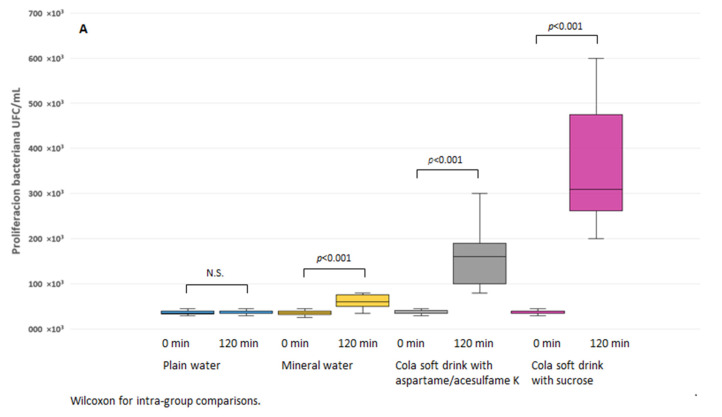
Bacterial proliferation of the basal dental biofilm and after the ingestion of the different types of beverages: (**A**) intra-group differences; (**B**) inter-group differences.

**Table 1 life-12-01776-t001:** General characteristics of the participants (*n* = 18).

Characteristics	*n*	%
Female	13	(72.2)
Consumption of 1–2 soft drinks per day *n* (%)	18	(100.0)
Tooth brushing 2 times a day *n* (%)	16	(88.9)
	median	IQR
Age (years)	17.0	(17.0–17.0)
DMFT	3.5	(2.0–4.0)

IQR: interquantile range; DMFT (Decayed, Missing and Filled Teeth).

## Data Availability

Supporting reported results can be found at https://drive.google.com/drive/folders/1d_1HgpcKEtQPmhMwOtco58yui3CWh7j3?usp=sharing (accessed on 12 April 2022).

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
