# Peer review of "Effects of Carbonated Beverage Consumption on Oral pH and Bacterial Proliferation in Adolescents: A Randomized Crossover Clinical Trial"

_life, 2022, doi:10.3390/life12111776_

Round 1
Reviewer 1 Report
1. figure no: 4 is complex and needs to be repeated using another type of charts.
2. you mentioned in the conclusion that the Ingestion of sucrose-containing soft drinks favors the acidification of salivary pH and the bacterial proliferation of dental biofilm. I think this is very basic fact. however you need to cast the light on the effect of the frequency of sucrose-containing soft drinks. so, my recommendation to re-write the conclusion in the way you show the strength of your study.
3. please you need to discuss the following statement in your discussion section:"Oral health educators can reinforce important practices such as decreasing the frequency of consumption and time duration of beverage contact with the teeth. "
4. Compare your study's findings with the following study:The association between beverage consumption pattern and dental problems in Iranian adolescents: a cross sectional study
Naimeh Hasheminejad 1 , Tayebeh Malek Mohammadi 2 , Mohammad Reza Mahmoodi 3 , Moein Barkam 1 , Arash Shahravan 4
Author Response
Suggestion 1.
Thanks for your recommendation.
Figure 4 has been modified for better understanding. Panel A) intragroup changes, panel B) intergroup changes (Lines 224-232).
Suggestion 2.
Thanks for the recommendation.
We adding the recommendation in the conclusion (Lines 323-325).
Suggestion 3.
We agree with the recommendation and add a paragraph to the discussion (Lines 310-318).
Suggestion 4.
Thanks for the recommendation. We added the article to the discussion (Lines 302-306).

Reviewer 2 Report
Dear Editor,
in my opinion the research is well designed and conclusions are relevant even if the sample is quite limited.
I suggest to Authors to reinforce the limitations of the research due to the limited sample.
Moreover I found a couple of articles that investigated the effect of sugar free drinks in biofilms and caries. Maybe they could be considered in the discussion.
Apart from these minor revisions I recommend for publication
Author Response
Suggestion 1. We rewrite about this limitation:
We recognize the small sample size as a limitation. This did not allow us to observe significant differences in changes in the pH of dental biofilm. Although each patient was his own control, it is necessary to include a greatest sample size for further studies (Line 293-296).
Suggestion 2.
Thanks for the recommendation.
We included two articles about this topic (Lines 280-282).
Giacaman RA, Pailahual V, Díaz-Garrido N. Cariogenicity induced by commercial carbonated beverages in an experimental biofilm-caries model. Eur J Dent. 2018;12(1):27–35.
Giacaman RA, Campos P, Muñoz-Sandoval C, Castro RJ. Cariogenic potential of commercial sweeteners in an experimental biofilm caries model on enamel. Arch Oral Biol. 2013 Sep;58(9):1116–22.
